# Sulfonylurea Use in Patients with Type 2 Diabetes and COPD: A Nationwide Population-Based Cohort Study

**DOI:** 10.3390/ijerph192215013

**Published:** 2022-11-15

**Authors:** Fu-Shun Yen, James Cheng-Chung Wei, Teng-Shun Yu, Chung Y. Hsu, Chih-Cheng Hsu, Chii-Min Hwu

**Affiliations:** 1Dr. Yen’s Clinic, No. 15, Shanying Road, Gueishan District, Taoyuan 33354, Taiwan; 2Institute of Medicine, Chung Shan Medical University, No. 110, Sec. 1, Jianguo N. Rd., South District, Taichung 40201, Taiwan; 3Department of Medicine, Chung Shan Medical University Hospital, No. 110, Sec. 1, Jianguo N. Rd., South District, Taichung 40201, Taiwan; 4Graduate Institute of Integrated Medicine, China Medical University, No. 91, Hsueh-Shih Road, Taichung 40402, Taiwan; 5Management Office for Health Data, China Medical University Hospital, 3F, No. 373-2, Jianxing Road, Taichung 40459, Taiwan; 6College of Medicine, China Medical University, No. 91, Xueshi Road, Taichung 40202, Taiwan; 7Graduate Institute of Biomedical Sciences, China Medical University, Taichung 40402, Taiwan; 8Institute of Population Health Sciences, National Health Research Institutes, 35 Keyan Road, Zhunan, Miaoli County 35053, Taiwan; 9Department of Health Services Administration, China Medical University, No. 91, Hsueh-Shih Road, Taichung 40402, Taiwan; 10Department of Family Medicine, Min-Sheng General Hospital, 168 ChingKuo Road, Taoyuan 33044, Taiwan; 11National Center for Geriatrics and Welfare Research, National Health Research Institutes, 35 Keyan Road, Zhunan, Miaoli County 35053, Taiwan; 12Faculty of Medicine, School of Medicine, National Yang-Ming Chiao Tung University, No. 155, Sec. 2, Linong Street, Taipei 11221, Taiwan; 13Section of Endocrinology and Metabolism, Department of Medicine, Taipei Veterans General Hospital, No. 201, Sec. 2, Shipai Road, Beitou District, Taipei 11217, Taiwan

**Keywords:** all-cause mortality, major adverse cardiovascular events, non-invasive positive pressure ventilation, invasive mechanical ventilation, bacterial pneumonia, lung cancer

## Abstract

We conducted this study to investigate the long-term outcomes of sulfonylurea (SU) use in patients with chronic obstructive pulmonary disease (COPD) and type 2 diabetes (T2D). We used propensity-score matching to identify 6008 pairs of SU users and nonusers from Taiwan’s National Health Insurance Research Database from 1 January 2000 to 31 December 2017. Cox proportional hazard models were used to compare the risks of mortality, cardiovascular events, non-invasive positive pressure ventilation, invasive mechanical ventilation, bacterial pneumonia, lung cancer, and hypoglycemia between SU users and nonusers. In the matched cohorts, the mean follow-up time for SU users and nonusers was 6.57 and 5.48 years, respectively. Compared with nonusers, SU users showed significantly lower risks of mortality [aHR 0.53(0.48–0.58)], cardiovascular events [aHR 0.88(0.81–0.96)], non-invasive positive pressure ventilation [aHR 0.74(0.6–0.92)], invasive mechanical ventilation [aHR 0.57(0.5–0.66)], and bacterial pneumonia [aHR 0.78(0.7–0.87)]. A longer cumulative duration of SU use was associated with a lower risk of these outcomes. This nationwide cohort study demonstrated that SU use was associated with significantly lower risks of cardiovascular events, ventilation use, bacterial pneumonia, and mortality in patients with COPD and T2D. SU may be a suitable option for diabetes management in these patients.

## 1. Introduction

Chronic obstructive pulmonary disease (COPD) is a heterogeneous pulmonary disorder with persistent airflow limitation, respiratory symptoms, and periodic exacerbation [1]. Inhaled particulate matter, physical inactivity, and aging are the main risk factors for COPD [2]. The prevalence of COPD has rapidly increased worldwide, probably due to industrialization, urbanization, and population aging [1,2]. The global prevalence of COPD among 30–79-year-olds was about 10.3% in 2019 [2], and the number of patients with COPD increased from 142 million in 1999 to 212 million in 2019 worldwide [3]. COPD is also a leading cause of death worldwide, accounting for the third leading cause of death in 2019 [1]. Patients with COPD are prone to type 2 diabetes (T2D) due to exposure to noxious particles, chronic inflammation, aging, and frequent corticosteroid use [4]. Diabetes mellitus shares common risk factors with COPD, and hyperglycemia can significantly decrease pulmonary function. Therefore, about 10% of patients with diabetes mellitus have COPD, and studies show that diabetes can worsen the progression and prognosis of COPD [4]. Both diabetes mellitus and COPD are linked to cardiovascular diseases [1,2,4]. However, few studies have attempted to identify which antidiabetic drug may interact with COPD and determine the most favorable therapeutic option in patients with COPD and T2D [4].

Sulfonylurea (SU) has been used as an antidiabetic since the 1950s [5]. One of the most used antidiabetic drugs worldwide [5], sulfonylurea shows a rapid onset of hypoglycemic activity and is inexpensive [5]. Although it may cause hypoglycemia, large clinical trials have proved the efficacy and safety of new-generation SUs and confirmed their place in diabetes management [5,6]. SU can trigger insulin secretion in a glucose-dependent manner, lower blood glucose levels, and partially improve insulin deficiency in T2D by binding to the sulfonylurea receptor (SUR) of the ATP-sensitive potassium (K_ATP_) channel and inducing channel closure [5,6]. In addition to the pancreas, K_ATP_ channels are also present in cardiac, skeletal, and smooth muscles [6]. Preclinical studies have shown that the opening and closing of potassium channels in airway smooth muscle and nerves may affect the contraction of the tracheobronchial tree [7]. However, no study has examined the effects of sulfonylureas on respiratory outcomes in patients with COPD. Therefore, this study aimed to investigate the long-term outcomes of SU use in patients with COPD and T2D.

## 2. Materials and Methods

### 2.1. Study Population

We identified patients from Taiwan’s National Health Insurance Research Database (NHIRD) (described in our previous study) [8]. Disease diagnosis in NHIRD was based on the International Classification of Diseases, Ninth and Tenth Revision, Clinical Modification (ICD-9/10-CM). The National Health Insurance administration randomly checks the records of inpatient and outpatient claims periodically to ensure accurate diagnosis and quality of care. The NHIRD links to the National Death Registry to obtain mortality information. The study was performed in accordance with the Declaration of Helsinki. The identifiable information of health care providers and patients was scrambled and deidentified before release to protect individual privacy. This study was approved by the Research Ethics Committee of China Medical University and Hospital (CMUH110-REC1-038-(CR1)). Informed consent was waived by the Research Ethics Committee.

### 2.2. Study Design

We identified patients diagnosed with COPD and T2D from 1 January 2000 to 31 December 2017. The diagnosis of COPD and T2D was based on the ICD codes (Appendix A) for at least two outpatient claims or one hospitalization. Previous studies have validated the ICD coding algorithms for the diagnosis of COPD and T2D with acceptable accuracy [9,10]. Exclusion criteria were as follows: (1) age < 40 or >80 years, (2) missing sex or age data, (3) diagnosis of type 1 diabetes, hepatic failure, or bacterial pneumonia or dialysis treatment before the index date, (4) lung cancers or death within 180 days after the index date to exclude latent diseases, (5) diagnosis of COPD or T2D before 1 January 2000, to exclude previous diseases.

### 2.3. Procedures

The day of concurrent diagnosis of COPD and T2D was defined as the comorbid date (Figure 1). Patients who received SU for more than 28 days after the comorbid date were defined as SU users, and those who did not receive SU before or after the comorbid date were SU nonusers. We defined the first day of SU use as the index date, and the index date for SU nonusers was defined as the same period from the comorbid date to the index date for SU users.

The following variables that could influence the results were considered and adjusted in this study: sex, age, smoking status, obesity (the diagnosis of overweight, obesity, or severe obesity), comorbidities (hypertension, dyslipidemia, peripheral arterial disease, chronic kidney disease, liver cirrhosis) diagnosed before the index date, prescriptions (item and number of antidiabetic drugs, item and number of antihypertensive drugs, statin, and aspirin), and duration of diabetes. We also calculated the Charlson Comorbidity Index (CCI) to evaluate disease burden [11] and determined the Diabetes Complication Severity Index (DCSI) score [12] to assess T2D complications.

### 2.4. Main Endpoints

We calculated and compared the incidence rate and hazard ratio of all-cause mortality, major adverse cardiovascular events (MACE; a composite outcome of coronary artery disease, stroke, and heart failure), hospitalization for COPD, noninvasive positive pressure ventilation (NIPPV), invasive mechanical ventilation (IMV) bacterial pneumonia, lung cancers, and severe hypoglycemia between SU users and nonusers. All-cause mortality was defined as discharge from hospitals with a mortality diagnosis (the cause of death confirmed with the National Death Registry). For the outcomes of interest, we censored patients on the date of mortality, respective outcomes, or at the end of the follow-up time on 31 December 2018, whichever appeared first.

### 2.5. Statistical Analysis

Propensity score matching was used for variables between SU users and nonusers [13]. Non-parsimonious multivariable logistic regression was used to estimate the propensity score for every patient with SU use as the dependent variate and the clinically related covariates as the independent variables, including sex, age, comorbidities, CCI, DCSI scores, medications, and duration of diabetes, as shown in Table 1. The nearest-neighbor algorithm was adopted to build matched pairs, assuming a standardized mean difference (SMD) of ≤0.10 as a negligible difference between the SU users and nonusers. Student’s *t*-test was utilized to determine the statistical difference in continuous variables (such as age), and the Chi-square test was utilized to test the statistical difference in categorical variables between study and control groups. We used crude and multivariable-adjusted Cox proportional hazard models to compare outcomes between SU users and nonusers. We adopted the Schoenfeld residuals to check the proportional hazard assumption. The results were presented as incidence rate (IR) per 1000 person-years (PY), hazard ratios (HRs), and 95% CIs between SU users and nonusers. We used the Kaplan–Meier method and log-rank test to describe and compare the cumulative incidences of all-cause mortality, IMV, and bacterial pneumonia over the follow-up time between SU users and nonusers. Stratified analysis was performed to assess the risks of death, NIPPV, IMV, and bacterial pneumonia among subgroups of SU users and nonusers. We calculated the risks of death, major adverse cardiovascular events, NIPPV, IMV, and bacterial pneumonia by three cumulative durations of SU use (28–499, 500–1799, ≥1800 days) relative to no-use to investigate the dose–response relationship.

A two-tailed *p*-value of less than 0.05 was considered statistically significant. SAS (version 9.4; SAS Institute, Cary, NC, USA) was used for the analyses of this study.

## 3. Results

### 3.1. Participants

From the National Health Insurance Research Database, 150,395 patients had T2D, 122,738 patients had COPD, 32,842 patients had coexisting COPD and T2D, and they were included in this cohort study (117,553 patients had T2D alone, 89,896 patients had COPD alone). After excluding ineligible patients, we identified 28,003 patients with coexisting COPD and T2D from the NHIRD from 1 January 2000 to 31 December 2017. Of these patients, 20,085 did not receive SU, and 7918 received SU. After 1:1 propensity score matching, we selected 6008 pairs of patients with or without SU use from this dataset. Figure 2 shows the flowchart of patient selection. The matched patients of SU users and nonusers were similar in the character of gender, age, smoking status, obesity, comorbidities, CCI, DCSI score, antidiabetic drugs, cardiovascular medications, and duration of diabetes. In the matched cohorts, the mean (SD) age of SU users and nonusers was 60.88 (8.66) and 61.02 (9.38) years, respectively; the follow-up time for SU users and nonusers was 6.57 and 5.48 years, respectively.

### 3.2. Main Endpoints

In the post-propensity score matching cohorts, 888 (14.78%) SU users and 1280 (21.30%) SU nonusers died during the follow-up period (incidence rate: 21.85 vs. 37.85 per 1000 patient-years). The multivariable-adjusted analysis showed that the aHR for SU users compared with nonusers was 0.53 (95% CI = 0.48–0.58, *p* < 0.001; Table 2). Compared to SU nonusers, SU users had significantly lower risks of major cardiovascular adverse events (aHR 0.88, 95% CI 0.81–0.96), noninvasive positive pressure ventilation (aHR 0.74, 95% CI 0.6–0.92), invasive mechanical ventilation (aHR 0.57, 95% CI 0.5–0.66), and bacterial pneumonia (aHR 0.78, 95% CI 0.7–0.87); SU users showed no significant difference for the risks of hospitalization for COPD (aHR 0.90, 95% CI 0.72–1.13), lung cancer (aHR 0.88, 95% CI 0.58–1.33), and severe hypoglycemia (aHR 0.91, 95% CI 0.73–1.13; Table 2).

Figure 3 depicts the cumulative incidences of invasive mechanical ventilation, bacterial pneumonia, and all-cause mortality between SU users and nonusers with the Kaplan–Meier method. SU users had significantly lower risks of invasive mechanical ventilation (log-rank *p*-value < 0.001), bacterial pneumonia (log-rank *p*-value = 0.001), and all-cause mortality (log-rank *p*-value < 0.001) compared with SU nonusers during the follow-up time.

### 3.3. Stratified Analysis

The stratified analysis of death between SU users and nonusers is shown in Appendix A. SU users had a significantly lower risk of death among all subgroups of patients except patients with obesity in this study compared to SU nonusers. A significantly lower risk of cardiovascular events was observed in SU users than in nonusers in the female and male sex, age 60 to 80 years, no obesity, no smoking, CCI ≥ 1, DCSI ≥ 2, OAD numbers = 2–3, and insulin use (Appendix A). A significantly lower risk of noninvasive positive pressure ventilation was observed in SU users than in nonusers in the female and male sex, age 70 to 80 years, no obesity, no smoking, DCSI ≥ 2, and OAD numbers = 2–3 (Appendix A). A significantly lower risk of invasive mechanical ventilation was observed in SU users than in nonusers in the female and male sex, age 50 to 80 years, no obesity, with or without smoking, CCI = 0 or ≥2, DCSI 0–≥2, OAD numbers = 0–3, with or without insulin use (Appendix A). A significantly lower risk of bacterial pneumonia was observed in SU users than in nonusers in the female and male sex, age 60 to 80 years, no obesity, no smoking, CCI = 0, DCSI ≥ 1, OAD numbers = 1 or >3, and no insulin use (Appendix A).

### 3.4. Cumulative Duration of SU Use

Compared to SU nonusers, patients who received SU for a cumulative duration of 500–1799 days (aHR 0.69, 95% CI 0.6–0.78), ≥1800 days (aHR 0.25, 95% CI 0.21–0.29) had a significantly lower risk of death. Patients who received SU for a cumulative duration of ≥1800 days had significantly lower risks of major adverse cardiovascular events (aHR 0.72, 95% CI 0.64–0.8), noninvasive positive pressure ventilation (aHR 0.45, 95% CI 0.33–0.62), invasive mechanical ventilation (aHR 0.33, 95% CI 0.27–0.4), and bacterial pneumonia (aHR 0.63, 95% CI 0.55–0.72) (Table 3).

## 4. Discussion

This nationwide population-based cohort study showed that SU use was associated with significantly lower risks of major adverse cardiovascular events, noninvasive positive pressure ventilation, invasive mechanical ventilation, bacterial pneumonia, and all-cause mortality than SU no-use in patients with COPD and T2D. A longer cumulative duration of SU use was associated with a lower risk of these outcomes.

Cardiovascular disease is the main comorbidity and cause of death in patients with COPD and T2D [1,2,14]. Previous studies have found that SUs increase the risk of cardiovascular diseases [15]. Reports show that glibenclamide may inhibit myocardial ischemic preconditioning and increase the risk of myocardial infarction [16]. However, subsequent large-scale clinical studies and randomized controlled trials showed no significantly higher risk of cardiovascular diseases with SUs [5]. Randomized controlled trials also showed no significant difference in the risk of cardiovascular diseases between SU and pioglitazone [17]. or linagliptin [18]. Our study revealed that SU use was associated with a significantly lower risk of composite major adverse cardiovascular events than SU no-use. A longer cumulative duration of SU use was associated with a lower risk of this outcome. The reduced risk of cardiovascular events among SU versus non-SU was prominent in patients with older age, higher disease burden, and more diabetes-related complications and severity. A 10-year follow-up analysis of the UK Prospective Diabetes Study demonstrated that intensive glucose control with SU or insulin was associated with a significantly lower risk of macrovascular complications [19]. Therefore, the reduction in cardiovascular events with SU use in this study may be due to the effective lowering of blood glucose levels.

COPD is an obstructive airway disease. Acute exacerbations often cause hypoxia and even respiratory failure [1,2]. Acute hypoxic respiratory failure may require endotracheal intubation for invasive mechanical ventilation. Studies have shown that noninvasive ventilation can decrease the risk of invasive ventilation and mortality in patients with hypoxia [1,20]. Invasive mechanical ventilation is required for patients with persistent hypercapnia and blood pH < 7.35 but >7.15 [21]. Our study showed that SU use was associated with a reduced risk of NIPPV and IMV in patients with COPD and T2D, and a longer cumulative duration of SU use was associated with a lower risk of these outcomes. SU may reduce the risk of hypoxia and respiratory failure in patients with COPD. Studies have shown that hyperglycemia can induce oxidative stress and pro-inflammatory cytokines, alter the structure of lung tissue and gas exchange, lead to respiratory muscle dysfunction, and decrease pulmonary function and physical endurance [4,22]. Therefore, the reduced risk of NIPPV and IMV may be attributed to the hypoglycemic effect of SU, to reduce oxidative stress, and improve lung function [4,22]. When sulfonylurea binds to SU receptor, it causes the closure of K_ATP_ channels, leads to the inhibited efflux of potassium ions, increased influx of sodium ions, cell membrane depolarization, the opening of voltage-gated calcium channels, and increased intracellular concentration of calcium ions; which can then cause the pancreatic β islet cells to release insulin, and the smooth muscle of the bronchus to contract [6]. Animal studies have shown that SUs may cause bronchoconstriction by the closure of K_ATP_ channels and worsen the clinical course of COPD [7]. However, the structure, size, physiology, and pathology of lungs between animals and humans have some differences [23,24]; the pharmacokinetics and pharmacodynamics of medications between animals and humans are also not identical [25,26]. Furthermore, our study showed no significantly higher risk of hospitalization for COPD with SU use. Wang et al. also showed that SU use was not associated with a higher risk of severe COPD exacerbation, and a longer cumulative duration of SU use was associated with a lower risk of this outcome [27]. Preclinical studies have shown that SU could mitigate the inflammatory reaction of K_ATP_ channel stimulation via mitogen-activated protein kinase (MAPK) pathways in macrophages and monocytes [28]. Studies show that tolbutamide can reverse the anoxic activation of K_ATP_ channels in the dorsal vagal neurons of the mouse brainstem [29]. Chronic intermittent hypoxia can result in insulin resistance and glucose intolerance [30]. SU may improve pulmonary function by decreasing hypoxia-induced glucose intolerance and insulin resistance. In brief, SU may ameliorate hyperglycemia and oxidative stress, reduce inflammation and insulin resistance, improve pulmonary function, and mitigate respiratory failure. More clinical studies are needed to realize the mechanisms of SU use and lung function change in patients with COPD.

More than 50% of patients with COPD have airway bacteria in the lower respiratory tract [31]. The frequent use of steroids and the airway retention of mucoid sputum may predispose patients with COPD to respiratory tract infections, exacerbation, and even bacterial pneumonia [2,4]. Diabetes mellitus may increase about two-fold the risk of bacterial pneumonia [8]. SUs were derived from sulfonamides, which were first used to treat typhoid fever in 1942 [5]. Furthermore, our study showed that SU could reduce the risk of bacterial pneumonia in patients with COPD and T2D. Clinical studies have shown that hyperglycemia may increase the glucose level in airway secretions and predispose patients to pulmonary infections [27,32]. This study showed a lower risk of bacterial pneumonia, which could be due to reduced airway glucose levels after SU use and decreased airway bacterial colonization.

COPD is a chronic lung disease with a high mortality rate. For several years it has been the third leading cause of death worldwide [1,2]. The main objective of COPD treatment is to mitigate disease progression and reduce exacerbations and even death [1,2]. Doctors should distinguish between dyspnea and exacerbation and prescribe adequate inhalers or medications according to the guidelines. Patients should also be encouraged to follow medical advice on medication use and discontinue smoking to reduce mortality risk [1,2]. Our study demonstrated that SUs were associated with a reduced risk of all-cause mortality in patients with COPD. A longer cumulative duration of SU use showed an association with a lower risk of death. Reduction in the risk of death with SU use was also significant in all subgroups of patients with COPD and T2D. The reduced risk of death from SU use in this study may be attributed to the decreased risk of cardiovascular events, ventilation use, and bacterial pneumonia.

The most worrying side effect of SU use is hypoglycemia. Clinical studies showed that about 1.8 % of patients receiving SUs had hypoglycemia per year, and the incidence of severe hypoglycemia was 0.86–5.6 per 1000 persons-years [5,6]. However, our study showed that SU was not associated with a significant difference in the risk of severe hypoglycemia. COPD and T2D are associated with a higher risk of lung cancer [22,33]. SU has been suspected to increase cancer risk because it promotes insulin secretion [34]. However, our study showed that SU use had no association with a significantly higher risk of lung cancer in patients with COPD and T2D.

Our study has several disadvantages. First, this administrative dataset lacks information on family history, dietary habits, physical activity, daily alcohol consumption, and smoking status, which may affect our comparison of the outcomes between SU users and nonusers. However, we used propensity score matching for 38 critical variables to align the health status of SU users and nonusers as much as possible. Second, this National Health Insurance database also lacks the results of biochemical and microbiological tests, hemoglobin A1C levels, and pulmonary function tests, precluding careful assessment of COPD and T2D severity. However, we used CCI, DCSI scores, the item and number of oral antidiabetic and antihypertensive drugs, insulin use, and the duration of T2D as a proxy for the severity of T2D. Third, this study mainly involved Taiwanese people with type 2 diabetes; therefore, these results may not apply to non-Taiwanese individuals or patients with non-T2D. Finally, there are always unknown, unmeasured confounding factors in a cohort study; randomized controlled trials are warranted to confirm our results. However, a well-designed real-world observational study can provide valuable information on clinical practice.

## 5. Conclusions

Our study showed that SU use was associated with lower risks of cardiovascular events, ventilation use, bacterial pneumonia, and all-cause mortality in patients with COPD and T2D. Perhaps, SU is a suitable option for diabetes management in patients with coexisting COPD and T2D.

## Figures and Tables

**Figure 1 ijerph-19-15013-f001:**
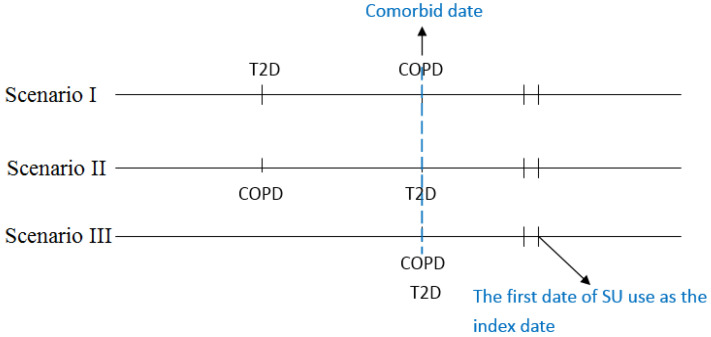
The scenarios for the diagnosis of T2D, COPD, comorbid COPD, and T2D, and the index date.

**Figure 2 ijerph-19-15013-f002:**
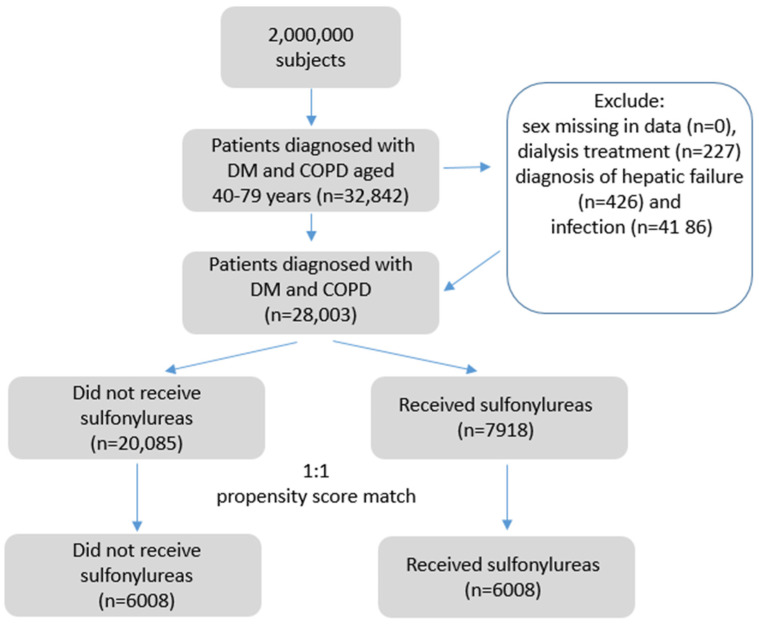
The flowchart of patient identification.

**Figure 3 ijerph-19-15013-f003:**
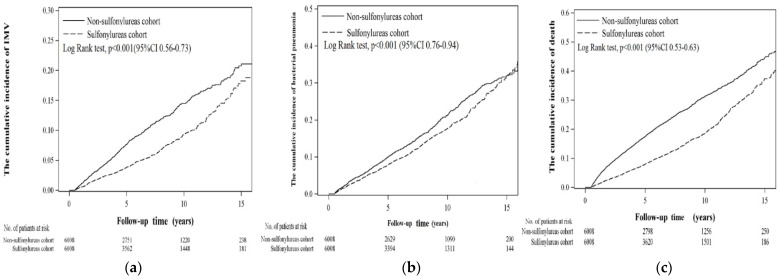
Cumulative incidence of (**a**) invasive mechanical ventilation (IMV), (**b**) bacterial pneumonia, and (**c**) death in sulfonylurea and non-sulfonylurea cohorts.

**Table 1 ijerph-19-15013-t001:** Baseline characteristics of patients with and without sulfonylureas.

Variables	Matched SU Nonusers		Matched SU Users		SMD *
	(*N* = 6008)		(*N* = 6008)		
	*n*	%	*n*	%	
Sex					0.014
female	2840	47.27	2882	47.97	
male	3168	52.73	3126	52.03	
Age					
40–49 ^1^	712	11.85	533	8.87	0.098
50–59	2060	34.29	2274	37.85	0.074
60–69	1938	32.26	2039	33.94	0.036
70–80	1298	21.60	1162	19.34	0.056
mean, (SD) ^2^	61.02	9.38	60.88	8.66	0.015
Comorbidities					
Obesity	153	2.55	139	2.31	0.015
Smoking	131	2.18	117	1.95	0.016
Hypertension	3842	63.95	3938	65.55	0.033
Dyslipidemia	3412	56.79	3477	57.87	0.022
Peripheral arterial disease	411	6.84	423	7.04	0.008
Chronic kidney disease	220	3.66	181	3.01	0.036
Liver cirrhosis	188	3.13	153	2.55	0.035
Charlson Comorbidity Index					
0	4291	71.42	4406	73.34	0.043
1	702	11.68	723	12.03	0.011
≥2	1015	16.89	879	14.63	0.062
Diabetes Complication Severity Index					
0	2322	38.65	2286	38.05	0.012
1	1199	19.96	1206	20.07	0.003
≥2	2487	41.39	2516	41.88	0.01
Medication					
Metformin	2845	47.35	2766	46.04	0.026
Dipeptidyl peptidase 4 inhibitors	450	7.49	419	6.97	0.02
Thiazolidinediones	246	4.09	184	3.06	0.056
Alpha-glucosidase inhibitors	393	6.54	351	5.84	0.029
Sodium-glucose cotransporter 2 inhibitors	13	0.22	13	0.22	<0.001
Insulins	2288	38.08	2226	37.05	0.021
Number of oral antidiabetic drugs					
0–1	4384	72.97	4429	73.72	0.017
2–3	1481	24.65	1509	25.12	0.011
>3	143	2.38	70	1.17	0.092
ACEI/ARB	3013	50.15	3041	50.62	0.009
β-blockers	2072	34.49	2049	34.10	0.008
Calcium-channel blockers	3514	58.49	3546	59.02	0.011
Potassium-sparing diuretics	911	15.16	936	15.58	0.012
Diuretics	2423	40.33	2453	40.83	0.01
Number of antihypertensive agents					
0–1	2453	40.83	2446	40.71	0.002
2–3	2455	40.86	2422	40.31	0.011
>3	1100	18.31	1140	18.97	0.017
Statin	2718	45.24	2727	45.39	0.003
Aspirin	2135	35.54	2143	35.67	0.003
Duration of diabetes					
Mean, (SD)	2.22	2.34	2.05	2.96	0.065

Abbreviations: SD, standard deviation; SMD, standardized mean difference; ACEI: angiotensin-converting enzyme inhibitors; ARB: angiotensin receptor blockers. ^1^: Chi-square test was used to test the statistical difference of categorical variables. ^2^: Student’s *t*-test was used to test the statistical difference of continuous variables. *: A standardized mean difference of ≤0.1 indicates a negligible difference between the two cohorts.

**Table 2 ijerph-19-15013-t002:** Outcomes between sulfonylurea versus non-sulfonylurea in patients with coexisting COPD and T2D.

Outcomes	Non-Sulfonylurea			Sulfonylurea								
	(*N* = 6008)			(*N* = 6008)								
	*n*	PY	IR	*n*	PY	IR	cHR	(95% CI)	*p*-Value	aHR ^†^	(95% CI)	*p*-Value
Death	1280	33,817	37.85	888	40,637	21.85	0.58	(0.53, 0.63)	<0.001	0.53	(0.48, 0.58)	<0.001
Major adverse cardiovascular events	972	30,577	31.79	1118	35,822	31.21	0.98	(0.9, 1.06)	0.5979	0.88	(0.81, 0.96)	0.0045
Hospitalization for COPD	145	33,449	4.34	181	40,051	4.52	1.04	(0.84, 1.3)	0.7021	0.90	(0.72, 1.13)	0.3544
NIPPV	180	33,573	5.36	173	40,303	4.29	0.81	(0.66, 1)	0.0458	0.74	(0.6, 0.92)	0.0067
IMV	505	33,313	15.16	389	40,027	9.72	0.64	(0.56, 0.73)	<0.001	0.57	(0.5, 0.66)	<0.001
Bacterial pneumonia	734	31,727	23.13	748	38,252	19.55	0.85	(0.76, 0.94)	0.0013	0.78	(0.7, 0.87)	<0.001
Lung cancer	44	33,758	1.30	48	40,575	1.18	0.90	(0.6, 1.36)	0.6167	0.88	(0.58, 1.33)	0.5435
Severe hypoglycemia	167	33,261	5.02	171	39,931	4.28	0.86	(0.69, 1.06)	0.1595	0.91	(0.73, 1.13)	0.3869

Abbreviations: COPD: chronic obstructive pulmonary disease; T2D, type 2 diabetes. PY: person-years; IR: incidence rate, per 1000 person-years; cHR, crude hazard ratio; aHR: adjusted hazard ratio; NIPPV, noninvasive positive pressure ventilation; IMV. invasive mechanical ventilation. aHR ^†^: multivariable analysis, including sex, age, comorbidities, and medications, as shown in Table 1.

**Table 3 ijerph-19-15013-t003:** The risk of different outcomes in patients with or without sulfonylurea use.

Variables	Death			Crude			Adjusted		
	Event	PY	IR	cHR	(95% CI)	*p*-Value	aHR ^†^	(95% CI)	*p*-Value
Non-use of SU	1280	33,817	37.85	1.00	(reference)	-	1.00	(reference)	-
Cumulative duration of SU (days)									
28–499	375	9164	40.92	1.14	(1.02, 1.28)	0.0224	0.90	(0.8, 1.02)	0.0908
500–1799	308	11,070	27.82	0.78	(0.69, 0.88)	<0.001	0.69	(0.6, 0.78)	<0.001
≥1800	205	20,403	10.05	0.25	(0.22, 0.29)	<0.001	0.25	(0.21, 0.29)	<0.001
Variables	Cardiovascular events				Crude		Adjusted		
	Event	PY	IR	cHR	(95% CI)	*p*-value	aHR	(95% CI)	*p*-value
Non-use of SU	972	30,577	31.79	1.00	(reference)	-	1.00	(reference)	-
Cumulative duration of SU (days)									
28–499	326	8016	40.67	1.29	(1.14, 1.47)	<0.001	1.05	(0.93, 1.2)	0.4188
500–1799	358	9781	36.60	1.14	(1.01, 1.29)	0.0374	1.01	(0.89, 1.14)	0.914
≥1800	434	18,025	24.08	0.75	(0.67, 0.84)	<0.001	0.72	(0.64, 0.8)	<0.001
Variables	Non-invasive positive pressure ventilation				Crude		Adjusted		
	Event	PY	IR	cHR	(95% CI)	*p*-value	aHR	(95% CI)	*p*-value
Non-use of SU	180	33,573	5.36	1.00	(reference)	-	1.00	(reference)	-
Cumulative duration of SU (days)									
28–499	59	9075	6.50	1.36	(1.01, 1.83)	0.0417	1.07	(0.79, 1.45)	0.6515
500–1799	60	10,943	5.48	1.18	(0.88, 1.59)	0.2657	1.03	(0.76, 1.38)	0.8647
≥1800	54	20,286	2.66	0.45	(0.33, 0.61)	<0.001	0.45	(0.33, 0.62)	<0.001
Variables	Invasive mechanical ventilation				Crude		Adjusted		
	Event	PY	IR	cHR	(95% CI)	*p*-value	aHR	(95% CI)	*p*-value
Non-use of SU	505	33,313	15.16	1.00	(reference)	-	1.00	(reference)	-
Cumulative duration of SU (days)									
28–499	137	8993	15.23	1.06	(0.87, 1.28)	0.5707	0.83	(0.69, 1.01)	0.0671
500–1799	142	10,837	13.10	0.90	(0.75, 1.09)	0.2931	0.80	(0.66, 0.96)	0.0198
≥1800	110	20,197	5.45	0.34	(0.28, 0.42)	<0.001	0.33	(0.27, 0.4)	<0.001
Variables	Bacterial pneumonia				Crude		Adjusted		
	Event	PY	IR	cHR	(95% CI)	*p*-value	aHR	(95% CI)	*p*-value
Non-use of SU	734	31,727	23.13	1.00	(reference)	-	1.00	(reference)	-
Cumulative duration of SU (days)									
28–499	205	8600	23.84	1.10	(0.94, 1.28)	0.2437	0.90	(0.77, 1.05)	0.1929
500–1799	239	10,391	23.00	1.07	(0.92, 1.24)	0.3747	0.97	(0.84, 1.12)	0.6722
≥1800	304	19,261	15.78	0.64	(0.56, 0.73)	<0.001	0.63	(0.55, 0.72)	<0.001

Abbreviations: PY: person-years; IR: incidence rate, per 1000 person-years; cHR, crude hazard ratio; aHR: adjusted hazard ratio. aHR ^†^: multivariable analysis, including sex, age, comorbidities, and medications, as shown in Table 1.

## Data Availability

Data of this study are available from the National Health Insurance Research Database (NHIRD) published by Taiwan National Health Insurance (NHI) Administration. The data utilized in this study cannot be made available in the paper, the Appendix A, or in a public repository due to the ‘‘Personal Information Protection Act’’ executed by the Taiwan government starting in 2012. Requests for data can be sent as a formal proposal to the NHIRD Office (https://dep.mohw.gov.tw/DOS/cp-2516-3591-113.html, accessed on 15 August 2022) or by email to stsung@mohw.gov.tw.

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
