# Peer review of "Sulfonylurea Use in Patients with Type 2 Diabetes and COPD: A Nationwide Population-Based Cohort Study"

_ijerph, 2022, doi:10.3390/ijerph192215013_

Round 1

Reviewer 1 Report

Line 116: Do you mean endpoints not „Main Outcomes“? Please correct.

Table 2: Please add an explanation of the abbreviations NIPPV, IMV, PY.

Please add the definition of PY: person-years, IR (incidence rate), cHR, crude hazard ratio; aHR: adjusted hazard ratio to the methods.

Can you add more information on the pharmacology of SU to the discussion?

Lines 244-248: Can you please explain why the use of SU reduces the risk of NIPPV and IMV in patients with COPD and T2D?

Line 251“However, the pathophysiology of animals is different from that of humans, and the pharmacological effects of SU in animals and humans may be dissimilar.” Can you explain the difference and why?

Line 103-105: “Patients who received SU for more than 28 days after the comorbid date were defined as SU users, and those who did not receive SU before or after the comorbid date were SU  nonusers.” This definition is not clear. What about patients who received SU for less than 28 days or before the comorbid date for any reason?

Study design

Line 92 “We identified patients diagnosed with COPD and T2D from January 1, 2000, to December 31, 2017.” Did all patients have COPD and T2D? What about patients who have COPD or T2D alone? Are they excluded from this study? If yes add this to the exclusion criteria. How many patients had COPD or T2D alone?

Conclusion:

„COPD and diabetes mellitus share the same risk factors and tend to occur together and exacerbate each other.” This was in the inclusion criteria to take patients diagnosed with COPD and T2D? What were the risk factors for each one? You didn’t study and compare patients who had COPD and diabetes mellitus.

Line 314: “However, few studies have examined the optimal therapeutic option for patients with coexisting COPD and T2D [2].” Please only include your conclusion in this section.

Author Response

Thank you for reviewing our manuscript and giving us insightful recommendations.

  1. Line 116: Do you mean endpoints not „Main Outcomes “? Please correct.

Response: Thank you for your recommendation. We have corrected the “Main Outcomes” to “Main Endpoints” on line 123 and 180.

  1. Table 2: Please add an explanation of the abbreviations NIPPV, IMV, PY. Please add the definition of PY: person-years, IR (incidence rate), cHR, crude hazard ratio; aHR: adjusted hazard ratio to the methods.

Response: We have added an explanation of the abbreviations NIPPV, IMV, PY on the footnote of Table 2, and added the definition of PY: person-years, IR (incidence rate), cHR, crude hazard ratio; aHR: adjusted hazard ratio to the section of Methods (page 3).

  1. Can you add more information on the pharmacology of SU to the discussion?

Response: We have added more information on the pharmacology of SU to the section of Discussion on page 12.

  1. Lines 244-248: Can you please explain why the use of SU reduces the risk of NIPPV and IMV in patients with COPD and T2D?

Response: We have explained the potential reasons for SU use reducing  risk of NIPPV and IMV in patients with COPD and T2D on page 12.

  1. Line 251“However, the pathophysiology of animals is different from that of humans, and the pharmacological effects of SU in animals and humans may be dissimilar.” Can you explain the difference and why?

Response: We have explained this difference and cited the references on page 12 as “However, the structure, size, physiology and pathology of lungs between animals and humans have some differences [23,24]; the pharmacokinetics and pharmacodynamics of medications between animals and humans are also not identical [25,26]”.

References:

  1. Harkema, J.R., Plopper, C.G., Pinkerton, K.E. Comparative Structure of the Respiratory Tract: Airway Architecture in Humans and Animals. In: Cohen, M.D., Zelikoff, J.T., Schlesinger, R.B. (eds) Pulmonary Immunotoxicology. Springer, Boston, MA, 2000; 1–59. doi:10.1007/978-1-4615-4535-4_1
  2. Williams K, Roman J. Studying human respiratory disease in animals--role of induced and naturally occurring models. J Pathol. 2016, 238, 220-32. doi: 10.1002/path.4658
  3. Lin JH. Species similarities and differences in pharmacokinetics. Drug Metab Dispos. 1995, ;23, 1008-21.
  4. Toutain PL, Ferran A, Bousquet-Mélou A. Species differences in pharmacokinetics and pharmacodynamics. Handb Exp Pharmacol. 2010, 199, 19-48. doi: 10.1007/978-3-642-10324-7_2

  1. Line 103-105: “Patients who received SU for more than 28 days after the comorbid date were defined as SU users, and those who did not receive SU before or after the comorbid date were SU nonusers.” This definition is not clear. What about patients who received SU for less than 28 days or before the comorbid date for any reason?

Response:  If a patient just takes SU for several days (such as <28 days), the drug use time may not be long enough to produce outcomes; furthermore, the investigated results may be illustrated by chance.

If we recruited patients who have used SU before the comorbid date, we may select prevalent SU users who were less likely to develop endpoints, which is called selection bias or prevalent user effect.

  1. Study design: Line 92 “We identified patients diagnosed with COPD and T2D from January 1, 2000, to December 31, 2017.” Did all patients have COPD and T2D? What about patients who have COPD or T2D alone? Are they excluded from this study? If yes add this to the exclusion criteria. How many patients had COPD or T2D alone?

Response: We have added the information of the participants on page 7 as” From the National Health Insurance Research Database, 150,395 patients had T2D, 122,738 patients had COPD, 32,842 patients had coexisting COPD and T2D, and they were included in this cohort study (117,553 patients had T2D alone, 89,896 patients had COPD alone)”.

  1. Conclusion: “COPD and diabetes mellitus share the same risk factors and tend to occur together and exacerbate each other.” This was in the inclusion criteria to take patients diagnosed with COPD and T2D? What were the risk factors for each one? You didn’t study and compare patients who had COPD and diabetes mellitus.

Response: We agree with your opinions and add the shared risk factors on page 14. Because COPD and T2D share the same risk factors and are prone to occur together, we investigated the long-term outcomes of SU use versus no-use in patients with coexisting COPD and T2D. 

  1. Line 314: “However, few studies have examined the optimal therapeutic option for patients with coexisting COPD and T2D [2].” Please only include your conclusion in this section.

Response: Thank you for kind recommendation. We have deleted this part on page 14.

Reviewer 2 Report

This study used propensity-score matching to examine the association between SU use and several long-term outcomes in patients with COPD and T2D. Authors found that SU used was associated with significantly lower risk of cardiovascular events, ventilation use, bacterial pneumonia, and mortality in these patients. 

This work is sound and well done. I only have a few comments regarding the methodology and reporting of results. 

1. Why excluding patients aged <40 or >80? Is there differences in this group? Please justify this sample selection.

2. Authors state that "Patients who received SU for more than 28 days after the comorbid date were defined as 103 SU users, and those who did not receive SU before or after the comorbid date were SU 104 nonusers". Were patients that did received SU before or after in less than 27 days since the comorbid date excluded?

3. What is the difference between comorbid date and index date? Which of those is considered study baseline? Please define more clearly study baseline and end of follow up.

4. How was age included in the propensity score matching and regression models? As a cuantitative or categorical variable? Please indicate so and include just the correct version in table 1.

5. Were assumptions checked? Particularly multicollinearity and the proportional hazards assumption. Please report.

6. Please indicate in the legend that table one refers to the matched analytic sample with the sample size.

7. Why use multivariable analyses when the covariables between groups in the matched analytic sample are already balanced? 

8. Please include confidence intervals and a risk table in the KM figures.

9. Please indicate sample size on all tables for each analysis. Specially in Table 2. 

Author Response

Thank you for your encouragement and giving us meaningful recommendations. We have rsponsed to the comments on the attahced file. 

Round 2

Reviewer 1 Report

Conclusion:

Please remove the sentence "COPD and diabetes mellitus share the same risk factors (physical inactivity and aging), and tend to occur together and exacerbate each other." Based on your cohort, which include patients with COPD and T2D, you can not make this conclusion. 

Author Response

Thank your for your encouragement and suggesions. We have remove this sentence from the consclusion on page 14.   
